# Silver@quercetin Nanoparticles with Aggregation-Induced Emission for Bioimaging In Vitro and In Vivo

**DOI:** 10.3390/ijms23137413

**Published:** 2022-07-03

**Authors:** Yuanyuan Li, Daming Xiao, Shujun Li, Zhijun Chen, Shouxin Liu, Jian Li

**Affiliations:** Key Laboratory of Bio-Based Material Science and Technology of Ministry of Education Northeast Forestry University, Harbin 150040, China; lyy2746@126.com (Y.L.); xiaodaming0607@gmail.com (D.X.); chenzhijun@nefu.edu.cn (Z.C.); liushouxin@126.com (S.L.); jianli@nefu.edu.cn (J.L.)

**Keywords:** quercetin, silver, self-assembling nanoparticles, aggregation-induced emission luminogens, bioimaging

## Abstract

Fluorescent materials based on aggregation-induced emission luminogens (AIEgens) have unique advantages for in situ and real-time monitoring of biomolecules and biological processes because of their high luminescence intensity and resistance to photobleaching. Unfortunately, many AIEgens require time-consuming and expensive syntheses, and the presence of residual toxic reagents reduces their biocompatibility. Herein, silver@quercetin nanoparticles (Ag@QCNPs), which have a clear core–shell structure, were prepared by redox reaction of quercetin (QC), a polyphenolic compound widely obtained from plants, including those used as foods, and silver ions. Ag@QCNPs show both aggregation-induced luminescence and the distinct plasma scattering of silver nanoparticles, as well as good resistance to photobleaching and biocompatibility. The Ag@QCNPs were successfully used for cytoplasmic labeling of living cells and for computerized tomography imaging in tumor-bearing mice, demonstrating their potential for clinical applications.

## 1. Introduction

Aggregation-induced emission (AIE) was first proposed by Tang and colleagues in 2001 [1]. Luminogens that produce aggregation-induced emission (AIEgens) are a better choice for light-up fluorescence imaging than conventional organic fluorophores, which always undergo aggregation-induced fluorescence quench [2,3,4]. AIEgens produce high intensity luminescence and have better resistance to photobleaching than conventional organic fluorophores [5,6]. These unique advantages offer the potential for in situ and real-time monitoring of biomolecules and biological processes [7,8]. However, the synthesis of AIEgens has typically been time-consuming and expensive, and the presence of residual toxic reagents in the process of synthesis reduces the biocompatibility of these materials. Thus, they are only used for bioimaging in cells, limiting their applications in vivo. Therefore, low-cost, biocompatible and green AIEgens are needed to advance the development of fluorescent materials.

Natural polyphenols have good biocompatibility and negligible biotoxicity [9]. Indeed, flavonols, which are the main category of flavonoids, are present in many fruits and vegetables and are thus considered to form part of a healthy diet [10,11]. In terms of fluorescence imaging, flavonols show excited state intramolecular proton transfer (ESIPT). Because of this, they have higher self-correction accuracy and larger Stokes shifts (>150 nm) than traditional fluorophores, which helps to reduce interference caused by self-absorption and self-fluorescence [12]. Although natural flavonols are particularly suitable as fluorescence-based sensors and biological imaging agents [13,14], research in such applications is still in its infancy. Fluorescent nanoparticles with a ‘core@shell’ structure can be formed between redox active AIEgens and silver ions via redox reactions, thus solving the problem of fluorescence quenching caused by the fusion of traditional fluorophores and plasma precious metals. AIEgens combined with precious metal nanoparticles have been successfully used in computerized tomography (CT) [15,16], providing a proof-of-concept for the development of flavonol-based AIE materials that can be integrated with silver nanoparticles.

The secondary metabolite quercetin (QC) is the most abundant flavonoid in plants [17] and can be extracted and separated from most vegetables, fruits and Chinese medicinal plants [18,19,20]. QC contains many phenolic hydroxyl groups with redox activity and is a good scavenger of free radicals [21,22]. Therefore, previous studies on the oxidation-reducing properties of QC mainly focused on its biological activity and synergistic effect. As a typical flavonol, QC is a natural AIEgen that demonstrates ESIPT [23,24]. Some previous studies have described QC-conjugated silver nanoparticles, but the focus of these studies was largely on improving the properties of the silver nanoparticles [25,26,27]. So far, the AIE properties of QC integrated with silver ions have not been described.

Here, we have devised a simple one-step strategy for the preparation of QC-coated silver nanoparticles (Ag@QCNPs) by redox reaction between QC and ammoniacal silver nitrate solution. The Ag@QCNPs have a clear core–shell structure and combine plasma scattering and strong aggregated-state fluorescence. The formation of a silver nucleus leads to aggregation and self-assembly of QC around it to form the core–shell nanostructures. The shell thickness, and thus fluorescence intensity, can be controlled by adjusting the amount of QC in the reaction mixture. The shell of QC reduces the cytotoxicity of the silver, allowing Ag@QCNPs to be successfully used for cytoplasmic and in vivo imaging (Figure 1).

## 2. Results

The main objective of this work was to develop a strategy to simplify the preparation of plasma fluorescent core–shell Ag@QCNPs in order to obtain biocompatible tissue-penetrant, polyphenol-based probes for bioimaging. The following procedure was adopted: Ag@QCNPs were obtained by one-step reaction of QC with ammoniacal silver nitrate solution;Redox reaction between QC and silver ions was confirmed by X-ray diffraction (XRD) and Fourier Transform Infrared (FT-IR) spectroscopy, and the core–shell structure of the nanoparticles was confirmed by transmission electron microscopy (TEM);The AIE and resistance to photobleaching of the Ag@QCNPs were determined by fluorescence excitation and emission spectra;The 3-(4,5-dimethyl-2-thiazolyl)-2,5-diphenyl-2H-tetrazolium bromide (MTT) assay was used to confirm that the Ag@QCNPs were not cytotoxic. They were then used for fluorescence imaging in living cells and for CT imaging of the tumor site in tumor-bearing mice. The Ag@QCNPs were restricted to the cytoplasm in living cells and were gradually cleared after accumulation at the tumor site in mice.

### 2.1. Characterization of Ag@QCNPs

#### 2.1.1. Redox Properties, XRD and FT-IR Characterization

Since the preparation of Ag@QCNPs is based on the redox properties of QC, cyclic voltammetry (CV) was used to investigate these properties. The cyclic voltammogram showed a reduction peak at 0.122 V and an oxidation peak at 0.292 V (Figure 1a), indicating that QC can reduce silver ions. QC was added to ammoniacal silver nitrate solution and the resulting precipitate was separated and washed with water to obtain Ag@QCNPs. The crystallinity of the Ag@QCNPs was analyzed by XRD. As shown in Figure 1b, the crystal peaks of Ag@QCNPs (red) corresponded well in both position and intensity to the silver standard card peaks (black) [28]. The FT-IR spectrum of Ag@QCNPs (Figure 1c) showed characteristic absorption peaks for -OH groups (3196 cm^−1^), benzene rings (1601–1502 cm^−1^), and C-O-C and CO groups (1360 cm^−1^, 1265 cm^−1^ and 1188 cm^−1^). Compared with the intensity of the -OH peak in the FT-IR spectrum of QC (Figure 1d), the intensity of the -OH peak in the spectrum of Ag@QCNPs was significantly reduced, indicating that a redox reaction had taken place between phenolic hydroxyl groups and silver ions.

#### 2.1.2. Morphology of Ag@QCNPs

TEM images showed that the Ag@QCNPs nanoparticles have a very obvious core–shell structure (Figure 2a,b). The silver core has a diameter of ~35 nm and is completely surrounded by a shell of QC. The high-resolution TEM image of an individual particle (Figure 2c) showed that the center of the nanoparticles is silver, with the crystal grain neatly arranged with a lattice spacing of 2.33 Å. The diffraction of the crystal can be seen from fast Fourier transform of the high-resolution TEM image (Figure 2d).

### 2.2. Fluorescence Characteristics of Ag@QCNPs

#### 2.2.1. Fluorescence Emission and Particle Size of Ag@QCNPs

Fluorescence emission spectra of Ag@QCNPs are shown in Figure 3a. According to our previous study on QC [24], the excitation wavelength was selected as 370 nm. The fluorescence spectrum of a solution of Ag@QC NPs in 100% tetrahydrofuran (THF) showed an emission peak at 480 nm. As the proportion of water in the solvent was gradually increased, the fluorescence intensity of the Ag@QCNPs increased steadily and the emission peak at 480 nm was gradually redshifted to 550 nm. This was similar to the phenomenon that the ketone emission of QC gradually increased with the increase of water ratio in THF/water solution reported in the literature [24]. The particle size of Ag@QCNPs dissolved in THF and THF/water (9:1 *v/v*) was analyzed by dynamic light scattering (DLS) (Figure 3b,c). The particle size of Ag@QCNPs in THF was about 30 nm, while Ag@QCNPs showed a small peak at ~30 nm and a much larger peak at ~110 nm in THF/water, corresponding to the monodispersed state and aggregated state of Ag@QCNPs, respectively. The commercially available fluorescent dye 4’,6-diamidino-2-phenylindole (DAPI) was used as a control when determining the resistance of the Ag@QCNPs to photobleaching. The fluorescence intensities of Ag@QCNPs and DAPI were recorded during irradiation with strong ultraviolet light (375 nm, 3 mW/cm^2^) for 100 min (Figure 3c). The fluorescence intensity of DAPI decreased by 20% during the experiment, whereas that of the Ag@QCNPs decreased by <10%, showing that Ag@QCNPs have good resistance to photobleaching.

#### 2.2.2. Shell Thickness and Fluorescence Intensity of Ag@QCNPs

The effect of the amount of QC in the Ag@QCNPs on their morphology was characterized by TEM. The shell thickness and fluorescence properties of the Ag@QCNPs could be adjusted by altering the amount of QC available for the redox reaction. Regardless of the amount of QC used in their preparation, all of the Ag@QCNPs were essentially spherical in shape (Figure 4a–d). When the amount of QC in the reaction mixture was increased from 75 mg to 375 mg, the thickness of the shell increased 4-fold to ~4 nm, the corresponding statistic shell thickness: 1.16 ± 0.21 nm (n = 124), 2.15 ± 0.26 nm (n = 133), 3.21 ± 0.31 nm (n = 120), and 3.95 ± 0.37 nm (n = 112). The fluorescence intensity of the Ag@QCNPs increased as the shell thickness of QC in the reaction mixture was increased (Figure 4e,f), showing that both the size and optical properties of the Ag@QCNPs can be adjusted by varying the reaction conditions.

### 2.3. In Vitro and In Vivo Imaging

#### 2.3.1. Cell Imaging

The MTT assay was used to determine whether the Ag@QCNPs show cytotoxicity. Human colorectal adenocarcinoma (HT-29) cells showed high survival (>95%) after incubating with concentrations of Ag@QCNPs as high as 400 μg/mL for 24 h (Figure 5a). Cell staining was investigated using HeLa cells. After co-incubating for 60 min with Ag@QCNPs (100 μg/mL), the nuclei of the cells were stained with DAPI. Images captured by confocal laser scanning microscopy (CLSM) clearly show that DAPI easily crosses the nuclear membrane to label DNA, whereas Ag@QCNPs are retained in the cytoplasm (Figure 5b). The merged images thus clearly show the entire cells.

#### 2.3.2. CT Imaging of Tumor Site in Mice

Six-week-old BALB/c nude mice were used to test the potential of Ag@QCNPs for CT imaging in vivo. Tumors were established by subcutaneous injection of S180 sarcoma cells into the left hips of the mice. Ag@QCNPs (1 mg/mL, 200 μL) were then injected via the tail vein and the intensities of the CT signals of the tumors in the left hips of the mice were recorded at different time intervals. Cross sections of the buttocks of one mouse, with the location of the tumor marked in red, are shown in Figure 6a. The CT signal intensities of the tumor site 0.5, 1, 2, 3, 4, 12 and 24 h after injection of the Ag@QCNPs are shown in Figure 6b. The accumulation of Ag@QCNPs in the tumor site was highest (128.6 HU) 2 h after injection, and then decreased gradually over the following 10 h.

## 3. Discussion

So far, research on flavonols has focused mainly on isolation and structure identification, pharmacological activity and signaling pathways and structure-activity relationship, as well as on the molecular mechanisms of interaction with biological macromolecules [29,30,31]. Although natural flavonols, which are biocompatible and show ESIPT, are particularly suitable as fluorescence-based sensors and biological imaging agents, there has been little research in this area. We have now used the flavonol QC to develop novel shell-core silver nanoparticles as fluorescent probes. Our new fluorescent probes have significant advantages over other synthetic AIE materials like 2,3-bis(4-(phenyl(4-(1,2,2-triphenylvinyl)phenyl)amino)phenyl)-fumaronitrile, an adduct of tetraphenylethene, triphenylamine, and fumaronitrile [16], since they have unique AIE characteristics and are easy to prepare, sustainable and biocompatible.

We first used cyclic voltammetry to determine the redox properties of QC. In agreement with literature reports, the oxidation peak of QC appeared at 0.292 V in the cyclic voltammogram, indicating that electrochemical oxidation of QC occurred on the surface of the electrode, corresponding to redox reaction of the adjacent 3′- and 4′-hydroxyl groups [32]. Since changes in pH affect the charge on QC, they have an important influence on the size, shape and morphology of the silver nanoparticles. Alkaline conditions, which accelerate deprotonation of QC, promote the redox reactivity of quercetin to some extent, and facilitate the formation of silver nanoparticles [33]. In the XRD spectra, the crystal diffraction peaks of the nanoparticles formed by reaction of QC with ammoniacal silver nitrate solution were consistent with the silver standard card. In the FT-IR spectra, the absorption peak of the hydroxyl groups in Ag@QCNPs was noticeably weaker than that of the hydroxyl groups in QC, indicating that a redox reaction had taken place between the phenolic hydroxyl groups of QC and silver ions [16,34,35]. TEM showed that the diameter of the silver nucleus was ~35 nm, which is close to the size of silver nanoparticles synthesized from plant extracts containing quercetin described in the literature [36]. After reaction of QC with the ammoniacal silver nitrate solution, an obvious shell-core structure was formed, with QC as the shell and silver as the core. This may be because alkaline solution promotes deprotonation of QC on the 3- and 7-positions, which would increase the electrostatic attraction between QC and silver ions. Hydrophobic interactions of QC also promote addition of more QC molecules onto the surface of the silver nanoparticles. These effects promote self-assembly of QC around the silver nucleus, thus forming the shell-core structure of Ag@QCNPs [16,32].

Having clarified the morphology and structure of the Ag@QCNPs, we next investigated their fluorescence characteristics and stability. The fluorescence spectrum of a solution of Ag@QCNPs in THF showed an emission peak at 480 nm. As the proportion of water in the solvent was increased, the emission peak at 480 nm was gradually redshifted to 550 nm. This phenomenon can be explained by the ESIPT that takes place in QC molecules that form the shell of the Ag@QCNPs. When the molecules are in a good solvent (THF), fluorescence is due to alcohol emissions but, as the volume fraction of the bad solvent (water) increases to 90%, molecular aggregation increases and intramolecular hydrogen bonds are gradually formed. Enol emissions therefore decrease and ketone emissions increase. This phenomenon can be summarized as aggregation-induced luminescence produced by ESIPT, and an AIE emission peak due to ketone emission [24]. This shows that the fluorescence of QC is not quenched when the molecules form a shell around the silver core and that the groups that contribute to the fluorescence properties of QC are retained. Ag@QCNPs showed a stronger signal peak at about 30 nm in THF, while the signal peak was weaker at 30 nm and stronger at about 110 nm in THF/water. The results suggested that Ag@QCNPs aggregated in THF/water. Aggregation of Ag@QCNPs in THF/water was mainly due to QC shells, and the aggregation of QC has AIE effect. Therefore, the signal strength of aggregated Ag@QCNPs in THF/water was stronger than that of individual particles, providing supplementary information about the AIE properties of Ag@QCNPs.

To be suitable for use as fluorescent probes, Ag@QCNPs must have stability in a specific environment in addition to showing AIE. The fluorescence intensity of Ag@QCNPs was attenuated to a smaller extent than that of DAPI, a commercially available fluorescent dye that is often used to fluorescently label cells, when irradiated with strong ultraviolet light, demonstrating that the Ag@QCNPs have good resistance to photobleaching. Since the AIE of Ag@QCNPs depends mainly on the QC in the shell, the amount of QC in the redox reaction has an important effect on the AIE properties of the Ag@QCNPs. The shell thickness and fluorescence intensity can be controlled by adjusting the amount of QC in the redox reaction, allowing greater flexibility and the design of Ag@QCNPs to fit specific needs. The fact that the shell thickness of the Ag@QCNPs can be increased only by increasing the amount of QC under fixed reaction conditions provides further evidence for self-assembly of QC around the silver core.

Before carrying out biological imaging, we investigated whether the Ag@QCNPs show any cytotoxicity. Even when incubated with Ag@QCNPs at concentrations as high as 400 μg/mL, the survival rate of HT-29 cells was >90%, indicating that Ag@QCNPs have good biocompatibility because of the protective effect of the QC shell. CLSM images showed that DAPI labeled the nucleus of HeLa cells, whereas Ag@QCNPs labeled the cytoplasm. Ag@QCNPs thus show good cellular uptake and accumulate in the cytoplasm, which means that they can be used to label subcellular structures [37]. When used for CT imaging in tumor-bearing mice, Ag@QCNPs were gradually removed from the tumor within 24 h, indicating little accumulation and demonstrating that Ag@QCNPs are safe and stable.

## 4. Materials and Methods

### 4.1. Material and Reagents

Quercetin (purity >98%), silver nitrate, sodium hydroxide, ammonia, THF and dimethyl sulfoxide (DMSO) were purchased from Shanghai Aladdin Biochemical Technology Co., Ltd. (Shanghai, China). DAPI was purchased from Beijing Suobao Biotechnology Co., Ltd. (Beijing, China). Human colorectal adenocarcinoma (HT-29) cells were purchased from Procell Life Science &Technology Co., Ltd., Wuhan, China. HeLa cells were purchased from Shanghai Honsun Biological Technology Co., Ltd., (Shanghai, China). Fetal bovine serum and penicillin-streptomycin were purchased from Thermo Fisher Scientific Co., Ltd., Shanghai, China. Deionized water was prepared using a Hitech-K Flow Water Purification System (Hitech Instrument Co., Ltd., Shanghai, China).

### 4.2. Preparation of Ag@QCNPs

In a typical experiment, aqueous ammonia was added dropwise to a mixture of aqueous silver nitrate (2 wt%, 10 mL) and aqueous sodium hydroxide (1 wt%, 10 mL) until the brown precipitate had completely disappeared, resulted in a solution of 0.06 M ammoniacal silver nitrate. QC was then added to the reaction mixture and the container was placed in a water bath at 70 °C for 4 h. The reaction mixture was centrifuged and the resulting precipitate was washed three times with deionized water. Finally, the precipitate was dispersed in deionized water, filtered through a 0.45 μm filter membrane and freeze-dried to provide Ag@QCNPs nanoparticles. Ag@QCNPs were prepared using different quantities of QC (75 mg, 125 mg, 255 mg and 375 mg).

### 4.3. Determination of Redox Properties of QC

A CHI 660e electrochemical workstation (Shanghai Chenhua Instrument Co., Ltd., Shanghai, China) was used to determine the redox properties of QC. The glassy carbon electrode was prepared as follows: (1) the electrode was polished with alumina polishing powder to remove surface residue; (2) the electrode was ultrasonicated for 10 min in (a) deionized water, (b) nitric acid/water (*v/v* 1:1) and (c) ethanol/water (*v/v* 1:1); (3) the electrode was activated, using 0.5 M H_2_SO_4_ as the electrolyte for cycle scanning, until the CV curve was smooth and stable. An ethanolic solution of QC was then dripped onto the surface of the electrode and allowed to dry naturally at room temperature before the test. A three-electrode system was used to determine the redox properties of QC: the working electrode was the glassy carbon electrode, the counter electrode was a platinum wire electrode, and the reference electrode was a silver/silver chloride electrode. The electrolyte was a mixture of KCl (1 M) and K_3_[Fe(CN)_6_] (5 mM). The scanning rate for the CV was set to 0.01 V/s, and the scanning potential was from −1 V to +1 V.

### 4.4. Characterization of Ag@QCNPs

The structure of the core–shell coating, morphology and electron diffraction of the Ag@QCNPs were characterized by transmission electron microscopy (TEM) using a JEM 2100 transmission electron microscope (JEOL Ltd., Tokyo, Japan). X-ray diffractograms of Ag@QCNPs were obtained using a D/max-2200VPC X–ray diffractometer (Rigaku Ltd., Tokyo, Japan), with a scan range of 30–85° (2θ). FT-IR spectra of QC and Ag@QCNPs were recorded using a Thermo Nicolet 10 FT-IR spectrophotometer (Thermo Fisher Scientific Co., Ltd., Beijing, China) between 500–4000 cm^−1^ wave numbers and at 4 cm^−1^ resolution.

### 4.5. Fluorescence Characteristics of Ag@QC NPs

Ag@QCNPs were dissolved in mixtures of THF/water containing different proportions of water (0%, 10%, 20%, 30%, 40%, 50%, 60%, 70%, 80% and 90%) and fluorescence emission curves were measured using an LS-55 fluorescence spectrometer (Perkin Elmer, Waltham, MA, USA). The concentration of Ag@QCNPs was 10 μg/mL and the excitation wavelength was 370 nm. The particle size distribution of the Ag@QCNPs was measured in THF/water (9:1 *v/v*) using a Zetasizer Nano ZS90 (Spectris China). The test temperature was 25 °C and the measured position was 0.85 mm. The commercially available fluorescent dye DAPI was used as a control to test resistance to photobleaching. Separate solutions of Ag@QCNPs and DAPI in THF (100 μg/mL) were irradiated with ultraviolet light and the fluorescence emission spectra of the two samples were recorded over 100 min to determine resistance to photobleaching. The fluorescence emission spectra of Ag@QCNPs prepared using 75 mg, 255 mg and 375 mg of QC, were measured in THF/water (9:1 *v/v*) to analyze the relationship between QC shell thickness and fluorescence intensity. The concentration of Ag@QCNPs was 10 μg/mL.

### 4.6. In Vitro and In Vivo Imaging

The biocompatibility of Ag@QCNPs was evaluated in HT-29 cells using the MTT assay. HT-29 cells were inoculated in a 96-well plate containing culture medium at a density of ~5000 cells per well. The cells were cultured at 37 °C for 24 h to allow attachment to the inner surface of the cover plate. Culture medium containing Ag@QCNPs (0, 12.5, 25, 50, 100, 200, 400 μg/mL) was then added to the wells (8 wells at each concentration) and the cells were incubated at 37 °C for 24 h. MTT solution (3 mg/mL, 50 μL) was added to each well, the cells were incubated at 37 °C for a further 4 h and DMSO (150 μL) was then added to each well. Absorbance at 490 nm was measured using an iMark microplate absorbance reader (BioRad, Tokyo, Japan). Cell viability was determined as the ratio of the absorbance of cells incubated with sample to that of cells incubated only with medium. Cell imaging was carried out by CLSM. HeLa cells (6000–7000 cells/mL) were inoculated into a 24-well plate and incubated at 37 °C for 24 h. Culture medium-containing Ag@QCNPs (10 μg/mL) was then added to each well and incubation was continued for 60 min. The cells were washed three times with PBS and the nuclei was stained with DAPI. Images was captured using a Ti-E A1 confocal laser microscope (Nikon, Japan), with an excitation wavelength of 375 nm and detection wavelengths of 410–460 nm and 500–600 nm, respectively for DAPI and Ag@QCNPs.

Six-week-old male BALB/c nude mice were used for in vivo imaging experiments. All animal experiments were carried out in accordance with the regulations of Henan Medical Research Center. Firstly, tumors were established by subcutaneous injection of S180 cells into the left hips of the mice. A solution of Ag@QCNPs (1 mg/mL, 200 μL) was then injected into the tail veins of the mice. The injected Ag@QCNPs were detected after 0.5, 1, 2, 3, 4, 12 and 24 h using a CT scanner (Siemens, Munich, Germany), at an accelerating voltage of 120 kV.

## 5. Conclusions

QC, which is found in many plants, is a very cheap natural raw material. We prepared Ag@QCNPs, which have a particle size of ~35 nm and a clear core–shell structure, by redox reaction of QC and silver ions. Ag@QCNPs show strong AIE, which can be tuned by altering the thickness of the QC shell, good resistance to photobleaching and good biocompatibility. The Ag@QCNPs were successfully used for both in vitro and in vivo imaging, indicating that they have the potential for clinical application.

## Data Availability

Not applicable.

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
