# Peer review of "Silver@quercetin Nanoparticles with Aggregation-Induced Emission for Bioimaging In Vitro and In Vivo"

_ijms, 2022, doi:10.3390/ijms23137413_

Round 1

Reviewer 1 Report

-          English improvement is required. Some examples but not all are as the following:

At 2.1.1, line 98, modify “The FTIR spectrum of Ag@QCNPs” with “The FT-IR spectrum of Ag@QCNPs”

At 4.4, line 295, modify “(Thermo Fisher Scientific (China) Co., Ltd.)” with “(Thermo Fisher Scientific Co., Ltd., China)”

-          At Introduction, line 33, the authors mentioned “has limited their applications”. Specify the applications.

-           Specify the concentration of ammoniacal silver nitrate solution.

-          At 3, line 190, authors mentioned “other synthetic AIE materials”. Specify the synthetic AIE materials.

-          At Fluorescence analysis, compared the obtained results with other data from literature.

Author Response

First of all, we would like to thank you for your positive and constructive comments and suggestions. Here, we have revised the manuscript by considering your suggestions. The followings are our response to each of the comments and the changes made in the manuscript.

  1. English improvement is required. Some examples but not all are as the following:

At 2.1.1, line 98, modify “The FTIR spectrum of Ag@QCNPs” with “The FT-IR spectrum of Ag@QCNPs”

Answer: Thank you very much for your suggestions. We have revised language and grammar of the manuscript. The representative changes about language and grammar are mainly in Introduction (lines 33-36 and 54-57). All words of “FTIR” have been substituted with “FT-IR”.

  1. At 4.4, line 295, modify “(Thermo Fisher Scientific (China) Co., Ltd.)” with “(Thermo Fisher Scientific Co., Ltd., China)”

Answer: Thank you very much for the suggestion. “(Thermo Fisher Scientific (China) Co., Ltd.)” has been modified to “(Thermo Fisher Scientific Co., Ltd., China) ” in line 313-314.

  1. At Introduction, line 33, the authors mentioned “has limited their applications”. Specify the applications.

Answer: Thank you very much for the suggestion. The synthesis of AIEgens has typically been time-consuming and expensive, and the presence of residual toxic reagents in the process of synthesis reduced the biocompatibility of these materials make them are only used for bioimaging in cells and limited their applications in vivo. For detailed revisions, please see lines 33-36.

  1. Specify the concentration of ammoniacal silver nitrate solution.

Answer: Thank you very much for the suggestion. According to the mass of each substance in the reaction, the concentration of ammoniacal silver nitrate was 0.06M. For detailed revisions, please see line 286-287.

  1. At 3, line 190, authors mentioned “other synthetic AIE materials”. Specify the synthetic AIE materials.

Answer: Thank you very much for the suggestion. An example of the synthetic AIE materials is provided, which is 2,3-bis(4-(phenyl(4-(1,2,2-triphenylvinyl)phenyl)amino) phenyl)-fumaronitrile, an adduct of tetraphenylethene, triphenylamine, and fumaronitrile. For detailed revisions, please see lines 203-205.

  1. At Fluorescence analysis, compared the obtained results with other data from literature.

Answer: Thank you very much for the suggestions. As the proportion of water in the solvent was gradually increased, the fluorescence intensity of the Ag@QCNPs increased steadily and the emission peak at 480 nm was gradually red shifted to 550 nm. This was similar to the phenomenon that the ketone emission of QC gradually increased with the increase of water ratio in THF/water solution reported in the literature. For detailed revisions, please see lines 126-128.

Reviewer 2 Report

This is a well-written and interesting paper. I have a few suggestions/questions, see below:

(1)    Line 127-128: I would like to see DLS performed at different aggregation states with varying ratios of THF/Water and direct comparisons to the emission spectra. From what I can tell in Figure 3b, this is only one individual DLS run.

(2)    Was an excitation scan done? How did you know that the excitation max was at 370 nm?

(3)    Figure 4f – how many samples were analyzed to acquire this data? I would like to see error bars on the fluorescence intensity. Likewise, how many particles were analyzed in the TEM images to determine shell thickness?

(4)    Figure 5 – there should be scale bars on these confocal images.

(5)    Figure 6a – I found these images to be a bit blurry. Perhaps it is just my copy, but please double check this.

(6)    Lines 227 – 228 – The Authors state that the signal strength of the aggregated particles is stronger than the signal for the individual particles by DLS. From Figure 3b, it appears as though DLS analysis was done by intensity value. As larger particles scatter more than smaller particles, is expected that the aggregated peak would be of higher intensity. The “number” value via DLS takes this into account. I would like to see DLS reported by both number and intensity.

Author Response

First of all, we would like to thank you for your positive and constructive comments and suggestions. Here, we have revised the manuscript by considering your suggestions. The followings are our response to each of the comments and the changes made in the manuscript.

(1)    Line 127-128: I would like to see DLS performed at different aggregation states with varying ratios of THF/Water and direct comparisons to the emission spectra. From what I can tell in Figure 3b, this is only one individual DLS run.

Answer: Thank you very much for your approval and suggestions on this manuscript. We measured the particle size of Ag@NPs in THF solution and compared it with that in THF/water (9:1 v:v) solution. According to the results of TEM, the particle size of Ag@NPs was about 35 nm, while that of Ag@NPs in THF solution was about 30 nm, indicating that Ag@NPs was uniformly dispersed in THF. The particle size of Ag@NPs in THF/water solution was mainly about 110 nm, indicating that Ag@NPs aggregated in THF/water. The surface of silver nanoparticle was the shell of QC, the AIE effect of QC in THF/water has been confirmed according to previous studies, so the particle size distribution of Ag@NPs in THF and THF/water can be used as a supplement to verify the AIE effect of Ag@NPs. For detailed revisions, please see lines 128-133 and Figure 3.

(2) Was an excitation scan done? How did you know that the excitation max was at 370 nm?

Answer: Thank you very much for the suggestion. Yes, an excitation scan was done. According to our previous study on the fluorescence characteristics of quercetin, when the excitation wavelength was about 370 nm, quercetin has a strong emission peak. For detailed revisions, please see line 121-122.

(3)  Figure 4f – how many samples were analyzed to acquire this data? I would like to see error bars on the fluorescence intensity. Likewise, how many particles were analyzed in the TEM images to determine shell thickness?

Answer: Thank you very much for the suggestion. The total amounts of quercetin for preparing the samples were 75, 125, 255 and 375 mg corresponding to the statistical shell thickness. The error bars and the numbers of particles analyzed in the TEM images to determine shell thickness were as follows: 1.16 ± 0.21 nm (n = 124), 2.15 ± 0.26 nm (n = 133), 3.21 ± 0.31 nm (n = 120), and 3.95 ± 0.37 nm (n = 112). For detailed revisions, please see lines 153-155. Error bars have also been supplemented in Figure 4f.

(4) Figure 5 – there should be scale bars on these confocal images.

Answer: Thank you very much for the suggestion. The scale bar of the confocal images has been supplemented in Figure 5.

(5) Figure 6a – I found these images to be a bit blurry. Perhaps it is just my copy, but please double check this.

Answer: Thank you very much for the suggestion. I have replaced Figure 6a with a better resolution.

(6) Lines 227 – 228 – The Authors state that the signal strength of the aggregated particles is stronger than the signal for the individual particles by DLS. From Figure 3b, it appears as though DLS analysis was done by intensity value. As larger particles scatter more than smaller particles, is expected that the aggregated peak would be of higher intensity. The “number” value via DLS takes this into account. I would like to see DLS reported by both number and intensity.

Answer: Thank you very much for the suggestion. Ag@QCNPs showed a stronger signal peak at about 30 nm in THF, while the signal peak was weaker at 30 nm and stronger at about 110 nm in THF/ water. The results suggested that Ag@QCNPs aggregated in THF/water. Aggregation of Ag@QCNPs in THF/ water was mainly due to QC shells, and the aggregation of QC has AIE effect. Therefore, the signal strength of aggregated Ag@QCNPs in THF/water was stronger than that of individual particles, providing supplementary information about the AIE properties of Ag@QCNPs. For detailed revisions, please see lines 242-245 and Figure 3.

Reviewer 3 Report

The creation of new fluorescent materials through the development of methods for the synthesis of low-cost, biocompatible and green agents is an urgent task. The authors present a simple method for obtaining core-shell nanoparticles in the form of silver particles coated with quercetin. The influence of synthesis conditions, in particular, the concentration of quercetin, on the product, which was characterized by a set of methods, was studied. A practical mapping experiment was also carried out using mice as an example, and it was shown that the created nanoparticles are suitable for this task. In general, the work makes a good impression, the manuscript is well structured, and the conclusions are backed up by experiment. There are just a few small remarks:

1.     Figure 4f should be rebuilt in the form of FL intensity (a.u.) vs Shell thickness (nm) instead of the dependencies on the concentration of the quercetin solution. It is the shell thickness that affects the fluorescence, and the concentration of the solution is not the true argument here.

2.     The authors write that they studied the created particles for photodegradation, comparing them with the well-known DAPI indicator. The manuscript does not specify the conditions for this experiment (only duration 100 min), in particular the power of the light source.

Author Response

First of all, we would like to thank you for your positive and constructive comments and suggestions. Here, we have revised the manuscript by considering your suggestions. The followings are our response to each of the comments and the changes made in the manuscript.

  1. Figure 4f should be rebuilt in the form of FL intensity (a.u.) vs Shell thickness (nm) instead of the dependencies on the concentration of the quercetin solution. It is the shell thickness that affects the fluorescence, and the concentration of the solution is not the true argument here.

Answer: Thank you very much for the suggestion. Figure 4f has been rebuilt in the form of FL strength (A.U.) vs Shell thickness (nm).

2,The authors write that they studied the created particles for photodegradation, comparing them with the well-known DAPI indicator. The manuscript does not specify the conditions for this experiment (only duration 100 min), in particular the power of the light source.

Answer: Thank you very much for the suggestions. The fluorescence intensities of Ag@QCNPs and DAPI were recorded during irradiation with strong ultraviolet light (375 nm, 3 mW/cm2) for 100 min. The wavelength and power of the light source have been supplemented in line 136.

Round 2

Reviewer 1 Report

Dear Sirs,

The manuscript was improved and it can be publish in this form.